# Early Childhood Learning Losses during COVID-19: Systematic Review

**Mustafa Uğraş [1], Erdal Zengin [1], Stamatis Papadakis [2,*] and Michail Kalogiannakis [2]**

1    Faculty of Education, Firat University, 23000 Elazig, Turkey
2    Department of Preschool Education, University of Crete, 74100 Rethymno, Greece
*    Correspondence: stpapadakis@uoc.gr

**Abstract:** The global education system has been significantly disrupted by COVID-19, and researchers are concerned with the impact this has had on students who have experienced learning loss. This study aims to systematically review the articles published in Science Direct, Wiley Online Library, SpringerLink, Sage Journals, Taylor & Francis Online, ERIC, JSTOR and Google Scholar on learning loss experienced by students in early childhood during the COVID-19 pandemic between 2020 and 2023. We conducted a systematic literature review of 33 articles published in the Web of Science (WOS), ERIC and Google Scholar electronic databases between 2020 and 2023. The review found a significant increase in early childhood learning losses. In addition, the present study found an increase in inequality, while certain demographic groups experienced more learning loss than others.

**Keywords:** COVID-19; early childhood education; learning loss; systematic literature review

## 1. Introduction

Global pandemic outbreaks in the 21st century have affected many regions [1–6]. Given how the COVID-19 virus spread around the globe, many measures were taken worldwide, including closing schools [7–11]. Face-to-face training was suspended to minimise the risk experienced during this period. As a result, the problem of how to meet the educational needs of the students arose [12–15]. To address this, educational institutions worldwide continued to move towards distance education [14,16,17]. However, the lack of adequate preparation of schools, administrators, teachers and parents [18] for this rapid and unplanned transition has been the subject of debate [19–22]. The first studies on students' academic performance during the pandemic found a decline in learning outcomes, expressed as learning losses [23–27]. Subsequent studies have found that the extent of learning loss is higher than indicated by initial findings [26,28].

Many factors have been identified as triggers of learning loss, such as holidays [29,30], earthquakes [31] and the global COVID-19 epidemic [32–34]. The global COVID-19 epidemic, especially from 2020 onwards, is thought to have caused an increase in learning losses [35]. To reduce the spread of COVID-19, social distancing strategies have come to the fore. As a result, face-to-face training activities have been suspended in countries affected by the pandemic. The costs of this decision include short-term learning losses experienced by students and the impact that has not yet been seen in their educational careers [33,36,37]. Around 1.6 billion students are estimated to be affected [38]. Educational researchers have explored the deficits, disruptions and learning losses in students' educational status [36,39,40]. The research found that while countries have moved to distance learning, they need help to prevent learning loss [33,41].

Learning is a cumulative and rapid process that provides children with the cognitive, social, emotional and physical developmental foundations upon which the rest of their life is built [42]. Learning helps children develop the social and emotional skills necessary to cope with the problems they may encounter in adulthood [42]. Cognitive development

is interrelated with and cannot be separated from social, emotional and psychomotor development in knowledge formation. Developing these skills in children will give them better chances at achieving an enhanced quality of life [42,43]. These aspects of learning and development are interrelated and mutually reinforcing in a sustainable process [42–44]. This helps children develop resilience to survive crises [43]. In early childhood, children experience a decline in their level of development if they fail to learn the basic skills they need to acquire during the critical period [45]. For these reasons, learning loss in early childhood is even more critical than at other levels of education. Even when schools open and education begins, the effects of learning loss do not stop [46]. These learning losses will impact children's educational attainment long-term, adversely affecting future generations.

In reviewing the studies conducted worldwide during the COVID-19 pandemic, it is striking that students were found to have high levels of learning loss [33,40,47]. Learning loss is a concept that affects all students in the educational process [39,48,49]. A pre-pandemic study by Black et al. (2017) found that an estimated 43% of all children under five low- and middle-income countries were at risk of not reaching their physical, socio-emotional and cognitive developmental potential [50]. In times of crisis, this rate is even higher. Research on the impact of previous pandemics and disasters (e.g., the Spanish flu in 1918 and the Asian flu pandemic in 1957) has shown that early childhood adversity is associated with later learning, behaviour and physical and psychological impairments in many children. Many long-term negative consequences are predicted [51]. Findings from studies of infants and toddlers during pandemics and epidemics have concluded that these children may be exposed to lifelong adverse effects, such as poor cognitive development [52] and lower educational attainment [53].

Economic instability and increased food insecurity during the pandemic had a more severe negative impact on young children's development due to increased caregiver stress and reduced health and social support [54]. A study by Lopez Boo et al. [55] found that due to this negative experience, young children were likely to experience significant losses in education, health, income and productivity throughout their lives [55]. Although they can cause significant losses and negatively affect students in this process, the adverse effects on the educational status of primary and secondary school children are emphasised more than the adverse effects seen in early childhood.

International research has revealed learning losses incurred from the first stages of the pandemic, except in some cases in early childhood [33,47,56–58]. These studies found that although learning losses are high among low-income and disadvantaged students, they are present in all segments [59,60]. A study conducted by Donnely and Patrinos (2022) concluded that students face the risk of losing 1/3 of the knowledge they had learned in one year [39]. The study found that a primary school pupil's absence from school for a year seriously impacts the grade in which he/she is placed and the subsequent educational process [47]. The loss of learning for a child in Year 3 is equivalent to a loss of learning at a rate of 2.8 per year up to year 10 [47]. The interruption of face-to-face education deprived students of the social support provided by the school and increased the number of positive COVID-19 cases and the time for which protective measures were implemented, and the uncertainty surrounding educational activities caused students to face negative outcomes [33]. Along with this process, being deprived of technological infrastructure and internet network [14,61], the education and economic level of families [62–65], the region where students live [66], the inability of young children to complete the educational process on their own [39] and the inability of students to interact with the school and the teacher [14] are among the other causes of learning loss.

To date, learning losses in early childhood and their long-term effects throughout life have not been widely studied [67,68]. Lopez Boo et al. [55] modelled the economic costs of early childhood school closures during the pandemic in 140 high-, middle- and low-income countries. This model shows that hundreds of millions of young children will likely experience significant lifelong learning losses due to the loss of early learning opportunities. Similarly, Bao et al. (2020) estimate that the disruption to learning in the United States due

to school closures during the pandemic resulted in a 31% decline in reading gains over nine months in 2020 [69]. While the short-, medium- and long-term impacts of the pandemic on households remain unclear, there is an urgent need for information to help monitor and mitigate its effects, particularly on children at risk [70].

A literature review has shown that there have been learning losses due to the COVID-19 pandemic and that these learning losses have reached significant proportions. The primary purpose of this research was to systematically analyse these studies to determine the learning loss caused by the closure of schools during the COVID-19 pandemic. To this end, a systematic analysis of the relevant articles in the literature completed between 2020 and 2022 was carried out. This study is expected to make three significant contributions to the literature:

- To provide a comprehensive review by synthesising the studies on the impact of the COVID-19 pandemic on the learning process of students;
- To answer the research question of whether the study of learning loss in early childhood during the COVID-19 pandemic is documented in the literature.

## 2. Aim of Research

This study aimed to systematically review the articles published in Science Direct, Wiley Online Library, SpringerLink, Sage Journals, Taylor & Francis Online, ERIC, JSTOR and Google Scholar on learning losses experienced by students in early childhood during the COVID-19 pandemic period between 2020 and 2023. For this purpose, answers to the following questions were sought:

1. What is the distribution of articles published in Science Direct, Wiley Online Library, SpringerLink, Sage Journals, Taylor & Francis Online, ERIC, JSTOR and Google Scholar on learning losses experienced by students in early childhood during the COVID-19 pandemic period of 2020–2023 according to year, study type, research method and design?
2. What are the research topics and results achieved in the articles published in Web of Science, ERIC and Google Scholar on learning losses experienced by students in early childhood during the COVID-19 pandemic between 2020 and 2022?

## 3. Method

A systematic review process [71], inspired by the methodology of Shute et al. (2017) and Fink (2019), was applied to gain a comprehensive understanding of learning loss in early childhood during the COVID-19 pandemic [72,73]. As mentioned above, learning loss in early childhood and its implications are remarkable. As no studies have systematically examined learning loss, this literature review will provide a general perspective on the extent and impact of learning loss in early childhood. A systematic examination of the results of studies on learning losses experienced in early childhood, especially during the COVID-19 pandemic, will contribute to future studies and serve as a reference source. This review followed the PRISMA (Preferred Reporting Items for Systematic Reviews and Meta-Analyses) guidelines [74] (please see Supplementary Materials for PRISMA checklist). The process followed was specific and included the following steps [75]:

- Specifying research questions;
- Searching databases;
- Inclusion/exclusion criteria;
- Selection of studies;
- Data analysis and extraction;
- Summary and interpretation of results;
- Writing the review report.

A publication classification form was used to analyse the articles in this study. While using the publication classification form, existing studies in the literature were used [39,76–79]. The content included descriptive information about the publication

classification form, year, publication type, research title, research method/design and research findings. The articles reviewed in the study were coded according to the themes (year, type of publication, research method, etc.) in the classification form. Thematic content analysis of the data was carried out, and subclassification and coding were performed. If more than one outcome was examined in the study, each outcome was coded separately. In this type of coding, the total data comprised the outcomes of the articles examined and analysis rather than the number of studies examined.

## 4. Research Strategy

We searched the Science Direct, Wiley Online Library, SpringerLink, Sage Journals, Taylor & Francis Online, ERIC and JSTOR databases for articles published between 2020 and 2023. Google Scholar ensured the research review included all significant published articles [80]. Thus, if an article was found in Scholar but not in the other databases, it was included in the research review. In addition, if the database offered filters for year of publication and language of publication, articles meeting the study's inclusion and exclusion criteria were specifically searched for.

According to Cronin et al. ([81], p. 41) [81,82], evaluating alternative terms with the same meaning increases the amount of information provided in a systematic literature review. Specific keywords were used to identify the studies included within this scope. Alternative keywords were identified using Boolean operators (AND, OR) from the database thesaurus. Terms used in the search string included core concepts consistent with our research topic and questions, such as "COVID", "learning loss" and "early childhood education", and synonyms, as shown in Table 1.

**Table 1.** Core concepts and synonyms.

| Core Concepts | Synonyms |
| --- | --- |
| COVID | COVID, COVID-19, Corona, SARS-CoV-2 |
| Learning loss | Learning loss, learning gap, learning gain, learning poverty, student performance, achievement gain, success gap, education gap, school effort |
| Early childhood education | Early childhood education, preschool education, kindergarten, primary education, elementary education. |

Boolean and simple operators were combined with brackets to create a search string. The search string was (COVID OR Corona OR "SARS-CoV-2" AND "learning loss" OR "learning gap" OR "learning gain" OR "learning poverty" OR "student performance" OR "achievement gain" OR "achievement gap" OR "education gap" OR "school effort" AND "early childhood" OR "kindergarten" OR "preschool"). The review looked at hundreds of articles. However, most referred to hypothetical or predicted learning loss. Article abstracts were scanned to reduce the number of articles to be reviewed. Studies were included if they analysed the loss of learning of students due to the closure of schools during the COVID-19 pandemic and if they reported on the effects (positive, negative or insignificant) on learning progress. After this screening process, there remained 33 articles. The data from these articles were tabulated and are part of the results section. Figure 1 shows the PRISMA process we followed.

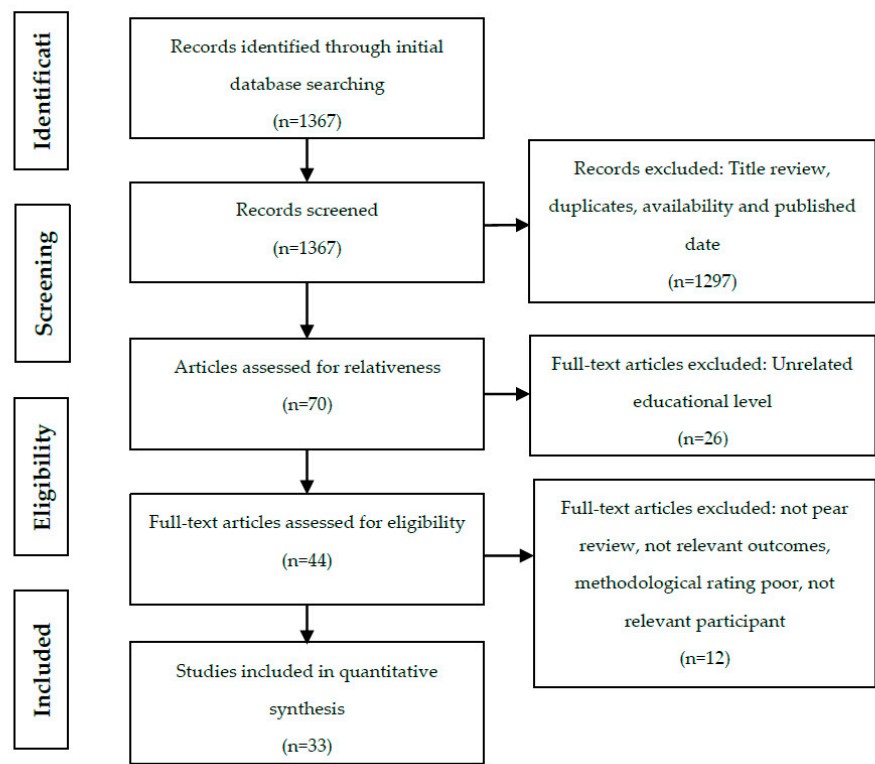

**Figure 1.** PRISMA review process.

## 5. Results

This section presents the analysis results of studies published in Science Direct, Wiley Online Library, SpringerLink, Sage Journals, Taylor & Francis Online, ERIC, JSTOR and Google Scholar between 2020 and 2023 on the topic of early childhood learning losses. The results are presented in terms of the inclusion and exclusion criteria of the studies in question, the distribution of the studies by year and type of publication, the number of publications identified in the databases examined and the preferred research method in the publications. Table 2 shows the criteria for including and excluding studies in this review.

**Table 2.** Inclusion/Exclusion Criteria.

| Inclusion Criteria | Exclusion Criteria |
|---|---|
| Articles published in a peer-reviewed journal | Extended abstracts, posters or presentations |
| Articles are written in English or Turkish | Articles are written in languages other than English or Turkish |
| Articles available as a full text | Articles where the full text is not accessible |
| Articles published in Science Direct, Wiley Online Library, SpringerLink, Sage Journals, Taylor & Francis Online, ERIC and JSTOR databases. | Articles published outside of Science Direct, Wiley Online Library, SpringerLink, Sage Journals, Taylor & Francis Online, ERIC and JSTOR databases. |
| Articles published during the COVID-19 pandemic (between 2020 and 2023) | The study must mention the loss of learning in early childhood education during COVID-19 (outside the years 2020–2023). |
| Articles of scientific value | Studies without scientific merit |
| Empirical studies in the field of early childhood education. | Studies on time periods outside of early childhood |
| Articles on learning loss | Articles not examining learning loss |

Table 2 shows the criteria for including and excluding studies in this review. According to the table, studies had to meet eight criteria to be included in the review. Similarly, seven criteria were set for excluding studies from the research.

Table 3 shows the databases in which the studies included in the research were scanned. Table 3 shows sixteen articles in Science Direct, two in Wiley Online Library, two in SpringerLink, four in Taylor & Francis Online, four in ERIC and five in Google Scholar.

**Table 3.** Data source and systematic review stages.

| Data Source | 1st Stage (Identification) | 2nd Stage (Screening) | 3rd Stage (Eligibility) | 4th Stage (Eligibility) | 5th Stage (Included) |
|---|---|---|---|---|---|
| Science Direct | 38 | 38 | 25 | 20 | 16 |
| Wiley Online Library | 2 | 2 | 4 | 3 | 2 |
| SpringerLink | 13 | 13 | 5 | 4 | 2 |
| Sage Journals | 11 | 11 | 4 | 2 | |
| Taylor & Francis Online | 12 | 12 | 6 | 5 | 4 |
| ERIC | 13 | 13 | 9 | 5 | 4 |
| JSTOR | 28 | 28 | 2 | 1 | |
| Google Scholar | 1250 | 1250 | 15 | 4 | 5 |

Table 4 shows the study distribution in the review by type of education, country, sample size, objective, study design and references. Looking at the studies published in 2020–2023 and included in the review, only one study was identified in Science Direct, Wiley Online Library, SpringerLink, Sage Journals, Taylor & Francis Online, ERIC, JSTOR and Google Scholar. It was found that nineteen studies were published in 2021 and included in the research, of which thirteen were found in the Web of Science, two in ERIC and three in Google Scholar. The number of studies published in 2022 and included in the research was thirteen, of which eight were found in the Web of Science, two in ERIC and three in Google Scholar. It can be seen that only one study published in 2023 was included in the research, and this was found in the Web of Science. In Table 4, in the type of education section, some studies included primary, secondary and higher education children as part of their sample. However, in this study, only learning loss data for students in grades 1–3 were considered.

Evidence of learning losses experienced by early childhood students during the COVID-19 pandemic is presented in Table 5. These observed learning losses were found to occur across different subjects, grade levels and geographical regions. These studies show the learning losses experienced by early childhood students during the COVID-19 pandemic upon removal from the face-to-face education system. It has been concluded that all the studies included in this review showed evidence of learning loss, regardless of the opportunities and learning levels of the students.

**Table 4.** Articles included in the systematic review.

| No. | Type of Education | Country/Region | Sample Size | Objective | Study Design | References |
|---|---|---|---|---|---|---|
| 1 | Primary school | United States | A study sample size of 3657 students was used. | This study examined the effect of school closures during COVID-19 on children's reading skills from different socio-economic backgrounds. | Quantitative research methods—survey | Bao, X., Qu, H., Zhang, R., and Hogan, T. P. (2020) [69] |
| 2 | Primary education | Netherlands | Primary school pupils in The Netherlands comprised the sample for this study. | This study assessed the impact of school closures on primary school performance using wealth data ($n \approx 350,000$) from The Netherlands. | Quantitative research methods-survey | Engzell, Frey and Verhagen (2021) [36] |
| 3 | Primary education and secondary education | Bangladesh, China, Japan, Korea and Indonesia | Primary and secondary English teachers in Asia were selected to participate. | Teachers' experiences during the pandemic were explored in this study. | Qualitative research methods—case study | Lee and Yin (2021) [82] |
| 4 | Early education | Spain | Children in year three to eight were included in the sample for this study. | This study aimed to identify the factors determining formal, non-formal and informal learning opportunities. | Quantitative research methods—survey | González and Bonal (2021) [83] |
| 5 | Primary education | Indonesia | This case study focused on Indonesian teachers in the first grade. | This study identified Indonesian first-grade teachers' emotions while teaching online during the pandemic. | Qualitative research methods—case study | Sanhadi Rahayu and Asanti (2021) [84] |
| 6 | Primary education | South Africa | Students in grades 2 and 4 of primary school were included in this study. | This study aimed to identify short-term reading learning losses in grades 2 and 4 during COVID-19. | Quantitative research methods— semi-experimental design | Ardington, Wills, and Kotze (2021) [85] |

**Table 4.** *Cont.*

| No. | Type of Education | Country/Region | Sample Size | Objective | Study Design | References |
|---|---|---|---|---|---|---|
| 7 | Early childhood | Ethiopia, Kenya, Liberia, Tanzania, and Uganda | Early childhood learners in Ethiopia, Kenya, Liberia, Tanzania and Uganda suffered learning losses following school closures due to the COVID-19 pandemic. | This study identified learning losses among young children living in Ethiopia, Kenya, Liberia, Tanzania and Uganda following the closure of schools due to the COVID-19 pandemic. | Quantitative research methods—survey | Angrist, N., de Barros, A., Bhula, R., Chakera, S., Cummiskey, C., DeStefano, J., and Stern, J. (2021) [47] |
| 8 | Primary education, secondary education and high school education | United Kingdom | Primary, secondary and high school students were included in this study. | This study determined the impact of a calibrated "educational production function" model. Predicted potential long-term losses in children's learning from the temporary shock of COVID-19-related school closures. | Quantitative research methods—survey | Kaffenberger (2021) [46] |
| 9 | Primary education, secondary education and high school education | Africa, Latin America and India | The survey was based on 388 school leaders. | This study explored what non-State Schools (NNSs) did to support distance learning, which might help them prepare for future events restricting education. | Quantitative research methods—survey | Cordeiro, Gluckman and Johnson (2021) [61] |
| 10 | Primary education | Turkey | The study group included first-grade teachers working in one of Turkey's Black Sea metropolises during the 2019–2020 and 2020–2021 academic years. | This study aimed to identify the difficulties faced by first-year primary school teachers. | Qualitative research methods—phenology | Uzun, Kar and Özdemir (2021) [86] |

**Table 4.** *Cont.*

| No. | Type of Education | Country/Region | Sample Size | Objective | Study Design | References |
|---|---|---|---|---|---|---|
| 11 | Primary education, secondary education and high school education | Turkey | The maximum diversity research sampling method was used, and 50 teachers from different sectors working at different levels of education were included. | This study identified teachers' views on the learning losses and solutions experienced by students due to the lack of face-to-face education disrupted by the COVID-19 pandemic. | Qualitative research methods—phenology | Baysal and Ocak (2021) [87] |
| 12 | Pre-primary and primary school | Greece | The sample surveyed consisted of 504 parents. | This study investigated the compliance of children aged 4 to 12 years with the COVID-19 closure restrictions and assessed the impact of school closure on children's educational, social, economic and psychological outcomes. | Quantitative research methods—survey | Siachpazidou, D. I., Kotsiou, O. S., Chatziparasidis, G., Papagiannis, D., Vavougios, G. D., Gogou, E., and Gourgoulianis, K. I. (2021) [88]. |
| 13 | Pre-primary school, primary school, secondary school, postsecondary school and tertiary levels | Bangladesh, Central African Republic, Chad, Congo D.R., Ghana, Guinea Bissau, Lesotho, Madagascar, Mongolia, Nepal, Pakistan (Punjab region), Sierra Leone and Zimbabwe. | Thirteen low- and lower-middle-income countries were included in the sample. | This study examined how disrupted schooling and dropout affect children's acquisition of basic skills before and during the COVID-19 pandemic. | Quantitative research methods—survey | Conto, C. A., Akseer, S., Dreesen, T., Kamei, A., Mizunoya, S., and Rigole, A. (2021) [89]. |
| 14 | Primary, secondary and higher education | Ethiopia, India, Peru and Vietnam | The sample for this research consisted of learners/students in primary education (5–11 years), secondary education (12–18 years) and higher education (19–26 years). | This study was conducted by Young Lives (YL) in Ethiopia, and two states in India, Peru and Vietnam to identify which socio-demographic groups benefit more from distance education. | Quantitative research methods—survey | Hossain, M (2021) [90] |

**Table 4.** *Cont.*

| No. | Type of Education | Country/Region | Sample Size | Objective | Study Design | References |
|---|---|---|---|---|---|---|
| 15 | Early childhood, primary school and secondary school | 624 teachers from across the world (mainly from Australia, New Zealand, Singapore and the USA) | The sample surveyed consisted of 624 teachers. | This study was conducted to explore teachers' experiences of the impact of COVID-19 in different countries. | Qualitative research methods—phenology | Phillips, L. G., Cain, M., Ritchie, J., Campbell, C., Davis, S., Brock, C., and Joosa, E. (2021) [91]. |
| 16 | Primary school | Norway | A total of 2453 first-grade students made up the sample for this study. | This study investigated the effect of the COVID-19 pandemic and emergency remote instruction on the writing of first-grade students in Norway. | Quantitative research methods—survey | Skar, G. B. U., Graham, S., and Huebner, A. (2021) [92]. |
| 17 | Primary school | New South Wales, Australia | A total of 3030 primary school pupils formed the sample for this study. | This study etermined the impact of educational disruption in the COVID-19 process on student success. | Quantitative research methods—survey | Gore, J., Fray, L., Miller, A., Harris, J., and Taggart, W. (2021) [56]. |
| 18 | Primary school | Indonesian | Eight primary school teachers formed the sample for this research. | This study examined the emotions of first-grade teachers in online classes during the pandemic. | Qualitative research methods—a case study design. | Arzaqi, R. N. and Romadona, N., F. (2021) [93]. |
| 19 | Primary school | Indonesian | Eight primary school teachers formed the sample for this research. | This study identified the challenges faced by teachers. | Qualitative research methods—case study design. | Sanhadi Rahayu, F. E., and Asanti, C. (2021) [84]. |
| 20 | Primary school | Swedish | A total of 97,073 primary school pupils (grades 1–3) were included in the sample for this study. | The learning losses of primary school pupils (grades 1–3) was examined in this study. | Quantitative research methods—LegiLexi data | Hallin, A. E., Danielsson, H., Nordström, T., and Fälth, L. (2022) [94]. |
| 21 | Primary school | Ghana | A total of 1844 primary school pupils formed the sample for this study. | This study examined learning experiences during the ten months of school closures in Ghana. | Quantitative research methods—survey | Wolf, S., Aurino, E., Suntheimer, N. M., Avornyo, E. A., Tsinigo, E., Jordan, J., and Behrman, J. R. (2022) [95]. |

**Table 4.** *Cont.*

| No. | Type of Education | Country/Region | Sample Size | Objective | Study Design | References |
|---|---|---|---|---|---|---|
| 22 | Preschool | US, California | The sample for this research consisted of four US preschool teachers (for 4-year-olds), four international preschool teachers (for 4- and 5-year-olds), three US kindergarten teachers (for 5-year-olds) and four US parents of 4- and 5-year-olds. | This study explored teachers' and parents' perspectives on the impact of physical/social distancing and school closure policies on children's socio-emotional development. | Qualitative research methods—semi-structured interview | Watts, R., and Pattnaik, J. (2022) [96]. |
| 23 | Primary education | Italy | The sample for this research was made up of children from primary schools in Italy. | This study investigated the impact of school closures during the pandemic on the mathematics skills of primary school children. | Quantitative research methods—survey | Contini et al. (2022) [97] |
| 24 | Early education | Spain | Approximately 250 primary school children participated in this study. | The representations young children use to express their knowledge of plant life were investigated in this study. | Mixed research methods | Zaballa, I., Merino, M., and Villarroel, J. D. (2022) [98]. |
| 25 | All levels of education | Mexico | All students studying in Mexico formed the sample for this study. | This study aimed to model the possible long-term effects of the epidemic on learning. | Quantitative research methods—Mexican education database. | Monroy-Gómez-Franco, L., Vélez-Grajales, R., and López-Calva, L. F. (2022) [66]. |
| 26 | Primary education | India | The sample consisted of private school teachers working in low-, medium- and high-budget private schools in two cities in India. | This study aimed to identify the challenges private school teachers face in low-, medium- and high-budget private schools in two cities in India. | Qualitative research methods—phenology | Pattnaik, J., Nath, N., and Nath, S. (2022) [14]. |

**Table 4.** *Cont.*

| No. | Type of Education | Country/Region | Sample Size | Objective | Study Design | References |
|-----|-------------------|----------------|-------------|-----------|--------------|------------|
| 27 | All levels of education | India | The study used secondary data sources, including the National Family Health Survey (2019–20) and the National Sample Survey (2017–18) and some recent reports, including the Lancet COVID-19 (2021) and Azim Premji Foundation Report (2021), to regenerate the relevant data for analysis. In addition, reports on learning loss (UNICEF, 2020) and ASER (Banerjee, 2020) were extensively reviewed. | This study aimed to identify the learning losses caused by school closures during COVID-19. | Quantitative research methods—survey | Singh (2022) [65] |
| 28 | Preschool education | Uruguay | Children aged 4–6 enrolled in kindergarten classes for 4- and 5-year-olds formed the sample for this study. | The heterogeneity of these effects across different socio-economic settings was explored in this study. | Quantitative research methods—survey | Odriozola-González, P., Planchuelo-Gómez, Á., Irurtia, M. J., and de Luis-García, R. (2022) [99] |
| 29 | Primary education | Belgium | The sample for this study consisted of primary school pupils in the Dutch-speaking Flemish region of Belgium. | This study evaluated the effects of school closures based on standardised tests in the last year of primary school in the Dutch-speaking Flemish region of Belgium. | Quantitative research methods—descriptive analysis | Maldonado, J. E., and De Witte, K. (2022) [57]. |
| 30 | Primary education | Turkey | Primary school classroom teachers were included in the sample for this study. | This study examined the learning losses experienced by primary school pupils in the distance learning process, in line with the opinions of classroom teachers. | Qualitative research method (case study)—content analysis | Sulak, S. E., and Çapanoğlu, A. Ş. (2022) [100]. |

**Table 4.** *Cont.*

| No. | Type of Education | Country/Region | Sample Size | Objective | Study Design | References |
|---|---|---|---|---|---|---|
| 31 | Primary education | Turkey | This study's sample included primary teachers. | This research aimed to discover teachers' opinions about the learning losses and solutions that are thought to have occurred during the COVID-19 pandemic. | Qualitative research method—content analysis | Tunç, Z., and Gök, B. (2022) [101]. |
| 32 | Primary education, secondary education and high school education | Turkey | The study group comprised 35 teachers working in Hatay, Turkey, during the 2021–2022 academic year. | This study aimed to identify the learning losses experienced by students in distance learning. | Qualitative research method (case study)—content analysis | Uyar, A., and Kadan, O. F. (2022) [102]. |
| 33 | Primary school | Netherlands | The sample for this research consisted of 886 students in grades 3–5. | This study examined the extent of the impact of the first school closure on vulnerable groups of pupils. | Quantitative, anonymised achievement data | Schuurman, T. M., Henrichs, L. F., Schuurman, N. K., Polderdijk, S., and Hornstra, L. (2023) [103]. |

**Table 5.** Characteristics of the included studies.

| Research Name | Authors | Database | Results of the Study |
|---|---|---|---|
| "As a Teacher, COVID-19 Means . . . ": Stories of How English Teachers in Asia Developed Resilience During the Pandemic | Lee, H., and Yin, J. (2021) [82] | Web of Science | The COVID-19 pandemic has led to several negative factors. These include lower academic standards, increased learning deficits and a lack of social skills. |
| COVID-19 school closures and cumulative disadvantage: Assessing the learning gap in formal, informal, and non-formal Education | González, S., and Bonal, X. (2021) [83] | Wiley Online Library | This study found that the learning loss of the advantaged and the disadvantaged students were at different levels in favour of the advantaged group. |
| Building back better to avert a learning catastrophe: Estimating learning loss from COVID-19 school shutdowns in Africa and facilitating short-term and long-term learning recovery | Angrist, N., de Barros, A., Bhula, R., Chakera, S., Cummiskey, C., DeStefano, J., and Stern, J. (2021) [47] | Science Direct | This study concluded that a child's learning loss in grade 3 could lead to a 2.8-year learning loss by grade 10. |

**Table 5.** *Cont.*

| Research Name | Authors | Database | Results of the Study |
|---|---|---|---|
| COVID-19 learning losses: Early grade reading in South Africa | Ardington, Wills and Kotze (2021) [85] | Science Direct | This study found short-term learning losses in reading for 2nd and 4th graders in South Africa. |
| Modelling the long-run learning impact of the COVID-19 learning shock: Actions to (more than) mitigate loss. | Kaffenberger, M. (2021) [46] | Science Direct | This study found a high rate of learning loss among students after schools were closed during the pandemic. |
| Emotional geographies of teaching online classes during COVID-19 pandemic: a case study of Indonesian first-grade elementary school teachers | Sanhadi Rahayu, F. E., and Asanti, C. (2021) [84] | Taylor & Francis Online | The research found that during the COVID-19 epidemic, problems such as teacher–student alienation and learning loss occurred. |
| Responses to COVID-19 From Non-State School Leaders in Latin America, Sub-Saharan Africa, and India: A Call for Educational Equity | Cordeiro, P. A., Gluckman, M., and Johnson, A. (2021) [61] | Web of Science | As a result of the research, it was found that the lack of infrastructure needs and the lack of internet access increase learning losses. |
| Learning loss due to school closures during the COVID-19 pandemic | Engzell, P., Frey, A., and Verhagen, M. D. (2021) [36] | ERIC | As a result of the research, it was found that the loss of learning during the closure of schools is approximately 3%, or 1/5 of an academic year, with values of 0.08 standard deviation. It was also found that the loss of learning was 60% for children from families with a low level of education. |
| Examining first-grade teachers' experiences and approaches regarding the impact of the COVID-19 pandemic on teaching and learning | Uzun, E. M., Kar, E. B., and Ozdemir, Y. (2021) [86] | ERIC | As a result of this research, it was found that the most critical problem in compulsory distance learning during the COVID-19 pandemic process was student learning loss. |
| Teachers' views on compensating learning losses caused during the COVID-19 pandemic. | Baysal, E. A., and Gürbüz, O. C. A. K. (2021) [87] | Google Scholar | This study found that the courses with the highest learning loss were Turkish, mathematics, science, physics, chemistry and English. |
| Action and Reaction of Pre-Primary and Primary School-Age Children to Restrictions during COVID-19 Pandemic in Greece | Siachpazidou, D. I., Kotsiou, O. S., Chatziparasidis, G., Papagiannis, D., Vavougios, G. D., Gogou, E., and Gourgoulianis, K. I. (2021) [88] | Web of Science | This study examined compliance with COVID-19 restrictions among children aged 4–12 and the impact of school closures on children's educational, social, economic and psychological outcomes. |
| The Kindergarten Headmaster's View of the Potential for Learning Loss in Early Childhood Education during Pandemic COVID-19 | Arzaqi, R. N., and Romadona, N., F. (2021) [93] | Google Scholar | This study aimed to identify learning losses caused by online and home-based learning during the COVID-19 epidemic. |

**Table 5.** *Cont.*

| Research Name | Authors | Database | Results of the Study |
|---|---|---|---|
| The effect of school closures on standardised student test outcomes | Maldonado, J. E., and De Witte, K. (2022) [57] | Wiley | The research found learning losses in reading, writing, mathematics and language learning. It was also found that learning losses were higher in schools with disadvantaged pupils. |
| Challenges to Remote Instruction During the Pandemic: A Qualitative Study with Primary Grade Teachers in India | Pattnaik, J., Nath, N., and Nath, S. (2022) [14] | Science Direct | The study found that primary school students' limited access to technological devices and the internet at home, limited amount of time spent with their parents, low educational attainment, unfamiliarity with the use of digital devices and limited participation in the distance learning process required by the school resulted in high rates of learning loss among students. |
| The potential effects of the COVID-19 pandlearningearnings | Monroy-Gómez-Franco, L., Vélez-Grajales, R., and López-Calva, L. F. (2022) [66] | Science Direct | The study found that a third of the information learned in a year is lost when students did not attend class in-person during the COVID-19 epidemic. It was also found that the learning loss of students studying in rural areas was more than three times that of students studying in central schools. |
| Children's drawing of plant life in the time of COVID-19: An analysis of the changes related to colour and two years-year period | Zaballa, I., Merino, M., and Villarroel, J. D. (2022) [98] | Web of Science | Due to the closure of schools during the COVID-19 pandemic, primary school children were found to have suffered severe learning losses. |
| Who Lost the Most? Mathematics Achievement during the COVID-19 Pandemic | Contini et al. (2022) [97] | Web of Science | The study found that children from lower-income and less-educated families in schools that closed during the COVID-19 epidemic experienced high rates of learning loss. In addition, it was found that the course in which students experienced the most significant loss of learning during the COVID-19 epidemic was mathematics. |
| School closures: Facing challenges of learning loss in india | Singh, C. B. (2022) [65] | ERIC | This study found that disadvantaged primary school children experience greater learning loss because their learning resources are minimal. This learning loss is more pronounced at the primary school level due to a lack of home resources and digital tools. |

**Table 5.** *Cont.*

| Research Name | Authors | Database | Results of the Study |
|---|---|---|---|
| School readiness losses during the COVID-19 outbreak. A comparison of two cohorts of young children | González et al. (2022) [99] | ERIC | During the COVID-19 pandemic, toddlers showed learning losses in motor and cognitive development and impaired attitudes to learning and internalising behaviour (range 0.13–0.27 SD). For students with a low socio-economic status, the learning losses were more pronounced. |
| Examining the learning losses experienced in the distance education process in line with the opinions of classroom teachers. | Sulak, S. E., Çetinkaya, S., and ÇAPANOĞLU, A. Ş. (2022) [100] | Google Scholar | During the COVID-19 pandemic, it was found that the students who experienced the most significant loss of learning were those in the mathematics and Turkish language courses. |
| Examination of the opinions of classroom teachers about learning losses that occurred during the COVID-19 pandemic. | Tunç, Z., and Gök, B. (2022) [101] | Google Scholar | It is noted that students may have experienced learning losses due to some psychological situations they experienced during the COVID-19 pandemic, the fear of losing loved ones, the fear of being ill, stress, domestic unrest and difficulties in accessing distance learning. |
| Teachers' Opinions on Students' Learning Losses During the COVID-19 Pandemic: A Case Study | Uyar, A., and Kadan, O. F. (2022) [102] | Google Scholar | The research found that students generally experienced learning losses in all courses. Another conclusion of the study was that Web 2.0 tools should be used, internet access should be provided, equipment support should be provided, infrastructure problems should be solved, fun activities should be added, and materials should be used to prevent learning losses in distance learning. |
| Potential effects of COVID-19 school closures on foundational skills and Country responses for mitigating learning loss | Conto, C. A., Akseer, S., Dreesen, T., Kamei, A., Mizunoya, S., and Rigole, A. (2021) [89] | Science Direct | This study found that skipping or dropping out of school was associated with lower literacy and numeracy scores. It was also concluded that countries are generally not doing enough to enable children to learn, prevent early school leaving or reduce learning loss, especially for marginalised and preschool children. |

**Table 5.** *Cont.*

| Research Name | Authors | Database | Results of the Study |
|---|---|---|---|
| Unequal experience of COVID-induced remote schooling in four developing countries | Hossain, M. (2021) [90] | Science Direct | This study found that students from more affluent families, urban areas and all countries with internet access were more likely to undertake distance learning. |
| No learning loss in Sweden during the pandemic: Evidence from primary school reading assessments | Hallin, A. E., Danielsson, H., Nordström, T., and Fälth, L. (2022) [94] | Science Direct | The study concluded that word decoding and reading comprehension scores were not lower than before the pandemic, that students with a low socio-economic status were not particularly affected, and that the proportion of students with poor decoding skills did not increase during the pandemic. These findings correlate with the Swedish education programme initiation during the pandemic. |
| Remote learning engagement and learning outcomes during school closures in Ghana | Wolf, S., Aurino, E., Suntheimer, N. M., Avornyo, E. A., Tsinigo, E., Jordan, J., and Behrman, J. R. (2022) [95] | Science Direct | This study found that there had been post-pandemic learning losses compared to pre-pandemic learning outcomes in Ghana. It also found that children with food insecurity and a low socio-economic status and those in public schools performed worse than their peers. |
| Perspectives of Parents and Teachers on the Impact of the COVID-19 Pandemic on Children's Socio-Emotional Well-Being | Watts, R., and Pattnaik, J. (2022) [96] | Springer | This study found that social deprivation, such as a lack of friendship, peer learning and communication, loss of playtime and lack of socialisation, affected children's socialisation skills, development of higher-order thinking, mental health and activity levels. In addition, their children exhibited externalising behaviours such as externalising, tantrums, seeking negative attention, aggression, lying and disrespect. |

**Table 5.** *Cont.*

| Research Name | Authors | Database | Results of the Study |
|---|---|---|---|
| The impact of COVID-19 on student learning in New South Wales primary schools: an empirical study | Gore, J., Fray, L., Miller, A., Harris, J., and Taggart, W. (2021) [56] | Springer | This study found no significant difference in student achievement growth between 2019 and 2020, as measured by the Advanced Mathematics and Reading Achievement Tests. However, the year 3 cohort in the least-advantaged schools (ICSEA < 950) experienced two-month-stunted growth in mathematics, while year 3 students in mid-ICSEA schools (950–1050) achieved two months of additional growth. No significant differences were found for Indigenous students or students in regional locations. |
| Learning Loss in Vulnerable Student Populations After the First COVID-19 School Closure in The Netherlands | Schuurman, T. M., Henrichs, L. F., Schuurman, N. K., Polderdijk, S., and Hornstra, L. (2023) [103] | Taylor & Francis Online | This study found that school closures caused a discontinuity in students' achievement growth on national standardised tests. It resulted in an average learning loss of 2.47 months in mathematics and 2.35 months in reading comprehension beyond the school closure period. |
| Surveying and resonating with teacher concerns during COVID-19 pandemic | Phillips, L. G., Cain, M., Ritchie, J., Campbell, C., Davis, S., Brock, C., and Joosa, E. (2021) [91] | Taylor & Francis Online | This study concluded that online learning, connecting/communicating with students and families, teaching quality and workload and valuing and investing in education and teacher professionalism address these issues. |
| Learning Loss During the COVID-19 Pandemic and the Impact of Emergency Remote Instruction on First Grade Students' Writing: A Natural Experiment | Skar, G. B. U., Graham, S., and Huebner, A. (2021) [92] | Science Direct | This study found that first-grade students during the pandemic had lower scores for writing quality, handwriting fluency and attitudes towards writing than their first-grade peers, who were tested one year earlier, before the onset of the COVID-19 pandemic. |

**Table 5.** *Cont.*

| Research Name | Authors | Database | Results of the Study |
|---|---|---|---|
| Emotional geographies of teaching online classes during COVID-19 pandemic: a case study of Indonesian first-grade elementary school teachers | Sanhadi Rahayu, F. E., and Asanti, C. (2021) [84] | Taylor & Francis Online | This study suggests that teachers experience negative emotions that lead them to emotional detachment when encountering problems such as uncooperative parents, excessive working hours, limited resources, teacher–student alienation, learning loss, demands of school rules and disagreements among colleagues. In addition, teachers admitted that having cooperative parents and supportive principals helps them to feel positive. |
| Modeling Reading Ability Gain in Kindergarten Children during COVID-19 School Closures | Bao, X., Qu, H., Zhang, R., and Hogan, T. P. (2020). [69] | Web of Science | According to this study, the rate of reading gain for kindergarten children during COVID-19 school closures without formal face-to-face instruction will decrease by 66% (2.46 vs. 7.17 points/100 days) compared to the business-as-usual scenario, resulting in a 31% lower reading gain from 1 January 2020 to 1 September 2020. |

## 6. Issues Caused by the COVID-19 Pandemic

There are many possible reasons for the learning losses experienced by students during the COVID-19 pandemic. Figure 2 shows a list of causes and the number of studies that reported causes. Family opportunities, educational support for children after the pandemic and economic factors caused learning losses in students [36,83,90,94,95,97,100,101]. In addition, it has been found that some psychological situations experienced by students during the COVID-19 pandemic may have caused learning losses due to the effects of the fear of losing loved ones, fear of being sick, stress, domestic unrest and difficulties in accessing distance education [101]. It was observed that in the existing studies in the literature, there are various suggestions for how the learning losses experienced can be compensated for [100]. It has been stated that to prevent learning losses in the distance education process, it is necessary to use Web 2.0 tools, provide internet access, provide equipment support, eliminate infrastructure problems, add fun activities and use materials [102]. It is also recommended that intensive learning programmes be implemented to reduce learning losses [46].

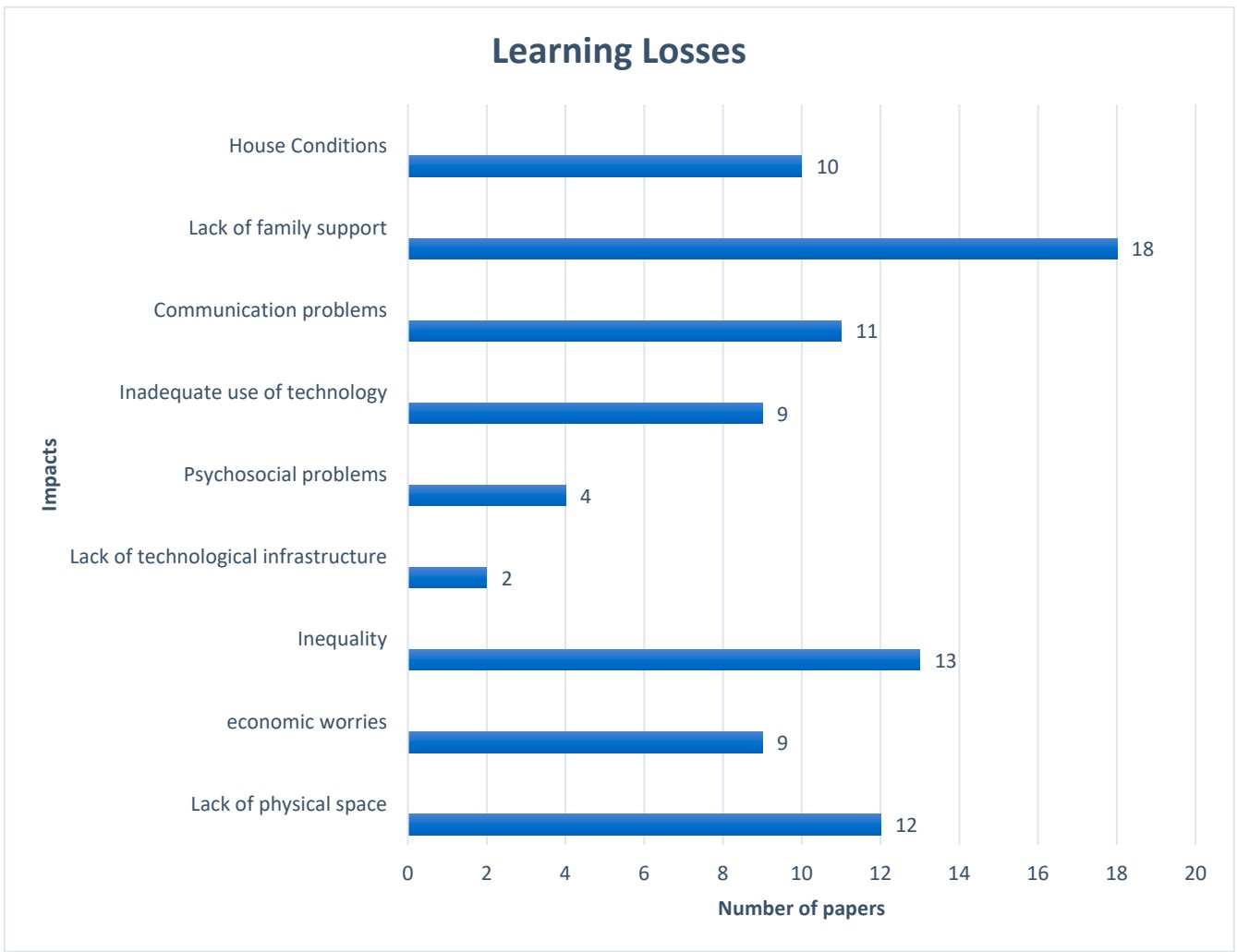

**Figure 2.** Reasons for the effects of COVID-19 on early childhood education were identified from the review articles (*n* = 33). Note: more than one reason may be discussed in some papers.

## 7. Impact of Early Childhood Learning Loss

Looking at studies of pandemic periods prior to the COVID-19 pandemic, it is clear that there will be both short- and long-term negative consequences for many children, who are particularly vulnerable in early childhood, when their brains are developing rapidly

and are very sensitive [51]. The first results on the impacts of the COVID-19 pandemic were obtained from parents and concerned the emergence of problems in the behaviour and mental state of children starting from the first wave of the pandemic [104]. These outcomes will adversely affect all early childhood age groups affected by the process. On the contrary, only some age groups are considered early childhood [54]. It is estimated that the learning problems experienced by children during the COVID-19 pandemic will have an impact for many years, especially in early childhood, when mental development is at a high level, and children are most receptive to learning [103]. The results of this literature review support this prediction [56,57,85,89,91,92,96,103]. Angrist et al. (2021) [47] found that one year of learning loss occurred during COVID-19, based on data from early childhood assessments in Ethiopia, Kenya, Liberia, Tanzania and Uganda. It was concluded that learning losses for a child in grade 3 could result in 2.8 years of learning loss by grade 10 [47]. However, according to research by Schuurman et al. (2021) [103], school closures due to the COVID-19 pandemic led to declines in students' academic performance, with a learning loss of 2.47 months in mathematics and 2.35 months in reading comprehension during school closures. These findings are not limited to these studies. Scientific studies in different countries [39], as well as current estimates and similar results [105], have shown that learning losses are severe in returning students in early childhood [106].

## 8. Comparison between Learning Losses in Early Childhood Education and Learning Losses in Other Education Levels Due to COVID-19

The level of learning loss experienced by early childhood students is high. For example, both Maldonado and De Witte (2022) [57] and Kuhfeld et al. (2020) [33] found that early childhood students experienced learning losses in certain subjects during the COVID-19 pandemic and were more affected than secondary school students. Similarly, Tomasik et al. (2021) [107] found that early childhood students were more affected than secondary students during the COVID-19 pandemic. While secondary school students were not found to have a loss of learning, early childhood students were. This is consistent with literature showing that early childhood students may be more vulnerable than secondary school students due to differences in developmental and cognitive abilities [39,84].

Similarly, a study conducted in Australia by Gore et al. (2021) [56] found that students in early childhood education in low-ICSEA schools were mainly more affected by the pandemic and experienced greater learning losses in mathematics and reading than those in other grades. It has been observed that these early childhood learning losses occur particularly in mathematics. The closure of schools due to the COVID-19 pandemic caused a decrease in the academic success of students in early childhood, with the closure of schools, and thus the discontinuation of mathematics courses [56,57,87,97,100], lasting an average of 2.47 months. This has been shown to cause a learning loss of 2.35 months in comprehension [103]. However, in a study found in the literature, from interviews with teachers, the courses for which students experienced the greatest learning loss were Turkish, mathematics, science, physics, chemistry and English.

On the other hand, students experienced less learning loss in social studies, biology, geography, history, music and art [87]. Again, a study on the learning losses experienced during the COVID-19 pandemic found that the students who experienced the greatest extent of learning loss were those taking mathematics, followed by Turkish [100]. Reports of learning losses are shown in Figure 3.

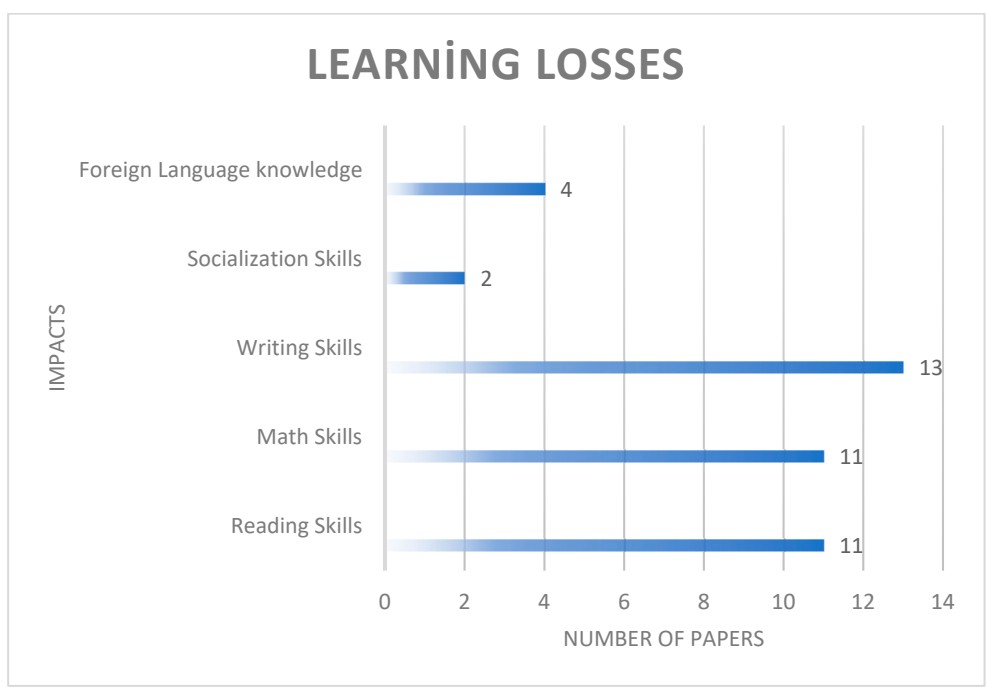

**Figure 3.** Early childhood education learning losses due to COVID-19 (*n* = 33). Note: more than one reason may be discussed in some papers.

## 9. Conclusions and Discussion

This research involved a systematic review of studies on learning losses in early childhood during the COVID-19 pandemic, published in the Web of Science, ERIC and Google Scholar databases, selected using specific criteria and published between 2020 and 2023. After reviewing the relevant databases, 33 studies were found to meet the previously established criteria. It was observed that the studies included in the review were mainly published between 2021 and 2023. As a result of the systematic review of these studies, it was found that there were significant learning losses in early childhood during the COVID-19 pandemic. Various reasons have been identified for these losses, and various proposals for their mitigation have been made.

Research on the impact of previous pandemics and disasters clearly shows that there will be both immediate and long-term negative consequences for many children, particularly those in early childhood, when brain development is still rapid and highly sensitive [51]. Early evidence during the COVID-19 pandemic shows an increase in behavioural problems in children since the pandemic, along with parents reporting mental health problems [104]. These impacts will adversely affect all early childhood age groups. On the contrary, only some age groups are considered early childhood [54]. It is estimated that the learning problems experienced by children during the COVID-19 pandemic will have an impact for many years, especially in early childhood, when mental development is at a high level and most open to learning [103]. The literature review results support this prediction [36,47,57,66,85,103]. Angrist et al. (2021) [47], using data from early childhood assessments in Ethiopia, Kenya, Liberia, Tanzania and Uganda, found a learning loss equivalent to one year during COVID-19. It was concluded that the learning loss for a child in grade 3 could reach 2.8 years by grade 10 [47]. The research conducted by Schuurman et al. (2023) [103] showed that the closure of schools due to the COVID-19 pandemic led to a decline in students' academic performance and a learning loss of 2.47 months in mathematics and 2.35 months in reading comprehension during the closure period. These findings are not limited to these studies. Scientific studies conducted in different countries [39], as well as current estimates and similar results [105], have shown that learning losses are severe for early childhood returnees [106].

Children removed from the face-to-face education system during the COVID-19 pandemic faced an educational process integrated into the distance education system. Along with this process, other factors considered included being deprived of technological infrastructure and the internet [14,61] and the education and economic level of families [64]; other causes of learning loss included the region [66], the inability of students in early childhood to carry out the educational process on their own [39] and the lack of interaction of students with the school and the teacher [14]. These situations, which cause the emergence of learning losses, have been identified in the studies found in the scanned databases. In the study conducted by Pattnaik, Nath and Nath (2022) [14], the primary school students included in the participant group had no access to technological devices at home and limited access to internet data; they found that the limited time spent by parents with their children, students' minimal level of education and unfamiliarity with the use of digital devices and the limited participation of children in the school's distance learning efforts caused profound learning loss among students. Furthermore, according to the study conducted by Engzell, Frey and Verhagen (2021) [36], children from families with a low level of education experienced a loss of 60% of school-learned knowledge. In the study conducted by Sanhadi Rahayu and Asanti (2021) [84], it was found that problems such as teacher–student alienation, learning loss, demands for school rules and disagreements among colleagues were encountered during the COVID-19 pandemic. Another study was carried out by Cordeiro, Gluckman and Johnson (2021) [61]. According to this study, infrastructure and system problems that prevent access to the internet increase learning loss.

The emergence of several negative factors, such as the decline in academic standards, the widening of the learning gap and the lack of social skills during the COVID-19 pandemic, have resulted in learning losses [82]. This is because COVID-19 disrupts daily life and leaves many families uncertain about the future (Glynn et al., 2021) [108]. Studies show that the COVID-19 pandemic has harmed the mental health of adults and their children, leading to learning losses in young children, and that these effects are widespread and lasting [109,110]. Given that there is some evidence that unpredictable situations in early childhood constitute an essential source of stress that may lead to impaired neurodevelopment and risk of psychopathology [108,111–113], it seems likely that this is a situation that negatively affects children's mental health [108]. Such psychological situations caused by the pandemic, including fear of losing loved ones in early childhood, fear of being sick, stress, domestic unrest and difficulties in accessing distance education, also cause student learning losses [101].

In general, the literature examining the impact of the learning process during the COVID-19 pandemic on early childhood students is limited in terms of the number of studies available, the geographical regions analysed and the number of students involved. This finding highlights the need for further research. Given that the issue of learning loss during and as a result of the pandemic is new, it is apparent why educational researchers have only just begun to investigate the learning losses experienced by students. However, more research is needed on how school closures affected student learning during the COVID-19 pandemic. In addition, the diversity of countries and regions where studies are conducted could be improved. Belgium, The Netherlands, Switzerland, Spain, the United States and Australia are examples of countries where studies have been conducted. Research that provides findings for a diverse range of countries and regions will fill the gap in the literature. Therefore, to obtain more accurate results, schools must continue investigating the learning loss associated with COVID-19 due to closures and distance learning strategies.

The importance of learning for children has been emphasised in many international documents. Article 28 of the United Nations Convention on the Rights of the Child [114] states that every child has the right to education. This is supported by the EU Charter of Fundamental Rights, Article 14, which states that everyone has the right to free access to education and training [115]. The school and home environments, and the relationship between the two, critically impact children's learning opportunities, achievements, motiva-

tion and confidence, especially in times of global crisis [116]. It is generally accepted that sudden changes in children's routines and learning styles are harmful (UNICEF, n.d.) [117].

Much of the literature that emerged during the global COVID-19 pandemic [118–124] revealed its adverse effects on children. This crisis situation affected children's participation in learning by triggering stress, lack of self-efficacy, fatigue and burnout. Both SDG 4.2., "Ensure that, by 2030, all girls and boys have access to a quality pre-primary education that prepares them for primary education", and SDG4., "Provide inclusive and equitable quality education and promote lifelong learning opportunities for all", are relevant in this context. UN member states plan to achieve these goals by 2030.

Many difficulties have been encountered in the early childhood education process, such as difficulty accessing online learning during the pandemic, lack of resources, lack of support from adults and lack of direct communication with teachers. These barriers are also barriers to achieving the Sustainable Development Goals. The extent of learning losses in early childhood shows that without aggressive policy measures, shocks to schooling and the economy will deepen the learning crisis. Children who are forced out of school may not return; those who do return will have lost critical periods for learning. In addition, the inequality between rich and poor families will widen as poor households are hit hard by the economic crisis.

Beyond the short-term effects of the pandemic, countries will experience significant long-term losses in human capital due to a lack of education. However, there is only one thing that needs to be done to reduce these negativities and turn them into long-term improvements: the realisation of quality education starting from early childhood. Quality education is critical to building a more inclusive and sustainable development process. From this perspective, if our focus is on sustainability, the future state of education depends on what we do in today's schools. Therefore, policies for equity and inclusive education are needed now more than ever to secure and strengthen the right to education in the context of achieving the UN Sustainable Development Goals. Starting with early childhood education, we should put quality education and sustainable development at the centre of learning. Students of today and tomorrow must be inspired to create new visions and paradigms to make this world a better place.

**Supplementary Materials:** The following supporting information can be downloaded at: https://www.mdpi.com/article/10.3390/su15076199/s1, please see supplementary materials for PRISMA checklist.

**Author Contributions:** Conceptualization, M.U. and E.Z.; methodology, M.U. and E.Z.; validation, M.U., E.Z., S.P. and M.K; formal analysis, M.U., E.Z., S.P. and M.K.; investigation, M.U., E.Z., S.P. and M.K.; writing—original draft preparation, M.U. and E.Z.; writing—review and editing, S.P.; visualization, M.U., E.Z., S.P. and M.K.; supervision, S.P. and M.K.; project administration, M.U., S.P. and M.K. All authors have read and agreed to the published version of the manuscript.

**Funding:** This research received no external funding.

**Institutional Review Board Statement:** Not applicable.

**Informed Consent Statement:** Not applicable.

**Data Availability Statement:** The data presented in this study are available on request from the corresponding author.

**Conflicts of Interest:** The authors declare no conflict of interest.

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
