# Peer review of "Early Childhood Learning Losses during COVID-19: Systematic Review"

_sustainability, doi:10.3390/su15076199_

Round 1

Reviewer 1 Report

The theme of this research, a literature review on the learning loss in early childhood education during the pandemic, is relevant and interesting. However this research has many limitations:

- early childhood education should comprise the period of learning that takes place from birth to 8 years old, Instead, some research refer to students of over the age of 8 and authors don't distinguish kindergarten from primary school.

- In lines 15-16 the authors say: "The study found increased considerably learning gains in early childhood." This sentence appears in contradiction with the overall meaning of the research.

- The sample sizes of students and their geographical locations are not mentioned; moreover, where evidence is claimed effect sizes are missing.

- 24 articles are too few.

Author Response

Dear Reviewer 1,

We are sincerely grateful for all your comments that improved our initial effort. In the revised manuscript, we tried to include all of your suggested changes We really appreciate your support throughout this process.

1/ Early childhood education should comprise the period of learning that takes place from birth to 8 years old, Instead, some research refer to students of over the age of 8 and authors don't distinguish kindergarten from primary school.

Thank you for your warning. Study outside of early childhood education was excluded. No distinction could be made because of the sampling dependence in the studies.

2/ In lines 15-16 the authors say: "The study found increased considerably learning gains in early childhood." This sentence appears in contradiction with the overall meaning of the research.

We made the necessary correction. Thank you very much. (Lines 16).

3/ The sample sizes of students and their geographical locations are not mentioned; moreover, where evidence is claimed effect sizes are missing.

We made the necessary correction. Thank you very much. (Lines 265).

4/ 24 articles are too few.

We expanded the keywords and date range. Accordingly, we included new studies. (Lines 240).

Reviewer 2 Report

Overall this systematic review has the potentials to become a valuable contribution to the field after the necessary amendments. Following are some of my notes and suggestions 

- It was quite difficult to follow the text flow due to syntactic weaknesses and inaccuracies. Text needs further English style and language editing.

- I am not sure if research question one is sufficiently answered. For example I miss the part where you describe the the distribution of the articles based on the study type, research method and design. This description, I believe, that would further strengthen your review's intention to identify the critical gap in the literature and to guide the future research

- It would be interesting to have a clearer depiction of the factors that may lead to the learning losses during the covid- 19 and the effects of the losses in children's life. Maybe if you could categorise them and present them in the form of a table following a description. 

- defending your argument regrading the 'Learning Losses Experienced in Early Childhood and Lessons Affected the Most' in Lines 264-265 & 287- 288 you are referring to children in 3rd grade. If I am not wrong 3rd grade belongs to primary school and not in Early Childhood Education.

Hope my notes will help strengthen your article.

  1.  

Author Response

Dear Reviewer 2,

We are sincerely grateful for all your comments that improved our initial effort. In the revised manuscript, we tried to include all of your suggested changes. We appreciate your support throughout this process.

1/ It was quite difficult to follow the text flow due to syntactic weaknesses and inaccuracies. Text needs further English style and language editing.

We proofread the document.

2/ I am not sure if research question one is sufficiently answered. For example I miss the part where you describe the the distribution of the articles based on the study type, research method and design. This description, I believe, that would further strengthen your review's intention to identify the critical gap in the literature and to guide the future research

We expanded the keywords and date range. Accordingly, we included new studies. (Lines240)

3/ It would be interesting to have a clearer depiction of the factors that may lead to the learning losses during the covid- 19 and the effects of the losses in children's life. Maybe if you could categorize them and present them in the form of a table following a description. 

Categorizations have been made. We thank you. (Lines 304-369)

4/ defending your argument regrading the 'Learning Losses Experienced in Early Childhood and Lessons Affected the Most' in Lines 264-265 & 287- 288 you are referring to children in 3rd grade. If I am not wrong 3rd grade belongs to primary school and not in Early Childhood Education. Hope my notes will help strengthen your article.

Thank you for your comment, studies outside of early childhood education was excluded

Reviewer 3 Report

This paper investigates the learning loss in the COVID-era for early-childhood students. Although the authors review enough papers, the main outcome of the paper is just a presentation of the reviewd papers. However, It is well-known  that COVID affected the educational process and primarily the youngest members of our community. As such, the paper does not commit something critically new.  

It would be desirable, to have a more thorough review that would investigate more critically the reasons, the different dimensions, the environment, the needs and the problems, and to compile and compare the results, presenting them (perhaps) diagrammatically. Moreover, an answer could be given on the possible ways to address those situations. 

On the other hand, what is the role of sustainability in this setup?  If we had a net loss in knowledge then nothing (and especially economic growth), is sustainable. Is there any sustainable solution? Aren't there any published papers on how to address those situations?

Thus, in my point of view this paper could go further, to investigate the qualitiative parameters of the reviewed papers and organize them critically.

Author Response

Dear Reviewer 3,

We are sincerely grateful for all your comments that improved our initial effort. In the revised manuscript, we tried to include all of your suggested changes. We appreciate your support throughout this process.

1/ This paper investigates the learning loss in the COVID-era for early-childhood students. Although the authors review enough papers, the main outcome of the paper is just a presentation of the reviewed papers. However, It is well-known that COVID affected the educational process and primarily the youngest members of our community. As such, the paper does not commit something critically new.  

Thank you for your comments. Arrangements have been made. We expanded the keywords and date range. Accordingly, we included new studies trying to commit a critically new text/

2/ It would be desirable, to have a more thorough review that would investigate more critically the reasons, the different dimensions, the environment, the needs and the problems, and to compile and compare the results, presenting them (perhaps) diagrammatically. Moreover, an answer could be given on the possible ways to address those situations. 

Categorizations have been made. We thank you. (Lines 304-369)

3/On the other hand, what is the role of sustainability in this setup? If we had a net loss in knowledge then nothing (and especially economic growth), is sustainable. Is there any sustainable solution? Aren't there any published papers on how to address those situations?

We tried to connect the scope of work and the relevant aim of sustainability. (Line 466-499).

Reviewer 4 Report

Dear authors, thank you for the opportunity to read your work. The topic is relevant. I would suggest expanding the results that include sample size, measurement, and main results in those 24 studies. Table 4 does not adequately present the data. It is possible to divide the table by the type of study: qualitative and quantitative, or more according to the criteria chosen. The data in the row should contain more information about researched relevant variables. 

Author Response

Dear Reviewer 4,

We are sincerely grateful for all your comments that improved our initial effort. In the revised manuscript, we tried to include all of your suggested changes. We appreciate your support throughout this process.

1/ I would suggest expanding the results that include sample size, measurement, and main results in those 24 studies.

We expanded the keywords and date range. Accordingly, we included new studies. (Lines240)

2/ Table 4 does not adequately present the data. It is possible to divide the table by the type of study: qualitative and quantitative, or more according to the criteria chosen.

Thank you. Arrangements have been made. (Lines 264)

3/ The data in the row should contain more information about researched relevant variables. 

Thank you. Arrangements have been made. (Lines 264)

Round 2

Reviewer 1 Report

The changes made have improved the quality of the paper.

Author Response

Dear reviewer, we would like to thank you for your comments.

In the attached file, you can find our detailed response.

With regards,

Reviewer 3 Report

 The paper has been improved and may be accepted in the present form.

Author Response

(The authors gave the same response as above.)
